# The Effects of Abdominal Hollowing and Bracing Maneuvers on Trunk Muscle Activity and Pelvic Rotation Angle during Leg Pull Front Pilates Exercise

**DOI:** 10.3390/healthcare11010060

**Published:** 2022-12-26

**Authors:** Eun-Joo Jung, Jae-Seop Oh

**Affiliations:** 1Center of Pilates BODY, No. 1105, 17 APEC-ro, Haeundae-gu, Busan 48060, Republic of Korea; 2Department of Physical Therapy, College of Healthcare Medical Science and Engineering, Inje University, 607 Inje-ro, Gimhae-si 50834, Republic of Korea

**Keywords:** Pilates exercises, abdominal hollowing, abdominal bracing, muscle activity, trunk muscles, pelvic rotation angle

## Abstract

Pilates methods use mats for trunk muscles stabilization exercises, and leg pull front (LPF) is one of the traditional Pilates mat exercises. Abdominal hollowing (AH) and Abdominal bracing (AB) maneuvers are recommended to stabilize the trunk muscles and prevent unwanted pelvic movement during motion. This study aimed to explore the effects of AH and AB on electromyography (EMG) activity of the trunk muscles and angle of pelvic rotation during LPF. A total of 20 healthy volunteers participated in the study. AH, AB, and without any condition (WC) were randomly performed during LPF exercise. Each was repeated three times for 5 s. The trunk muscle activities were measured using EMG and rotation of pelvis was measured using a Smart KEMA device. The activities of the transversus abdominis/obliquus internus abdominis (TrA/IO) and right obliquus externus abdominis (EO) muscles were highest in LPF-AH compared to the other conditions. Multifidus (MF) activity was significantly greater in LPF-AH and LPF-AB compared to that of without any condition. The pelvic rotation angle was significantly smaller in LPF-AB. Therefore, AH maneuver during LPF for trunk muscle stabilization exercises is suitable for selective activation of the TrA/IO, and AB maneuver during LPF is recommended for the prevention of unwanted pelvic rotation.

## 1. Introduction

Pilates is a popular form of exercise for conditioning and rehabilitation. It is used to increase stability, control exercise, and strengthen muscles [1,2]. As such, strengthening physical strength and recruitment improves balance and posture during activities of daily living (ADL). Several studies have investigated the utility of Pilates exercises for increasing trunk stability [3,4]. The leg pull front (LPF) is a popular choice among the traditional Pilates mat exercises. 

The LPF is clinically recommended because it challenges the trunk muscles to maintain correct abdominal and pelvic postures. The plank exercise provides optimal enhancement of abdominal muscle activity and strength [5]. Additional limb movement has been shown to enhance abdominal muscle activity during exercise [6]. Abdominal muscle activation occurs earlier and with a greater amplitude when predictable perturbation is applied via additional limb movement [7]; the muscle activation pattern is related to the direction of limb movement [6,7,8]. However, the LPF can cause unwanted pelvic movement during hip extension.

Comerford and Mottram [9] defined uncontrolled pelvic motion as faulty, repeated, and unwanted movements in the pelvic region. Uncontrolled pelvic motion can lead to excessive mechanical load on the lumbar spine; the repeated lower extremity exercises increase lumbar pelvic movement, which causes musculoskeletal disorders, including lumbopelvic instability [10,11,12]. To minimize lumbopelvic movement while moving the limbs, patients are trained to stabilize the trunk and pelvis using various methods.

To prevent such movement of the lumbar spine, therapists often ask patients to stabilize the trunk using abdominal hollowing (AH) and abdominal bracing (AB). In AH, selective contraction of the TrA and IO is achieved by pulling the navel inward towards the spinal column [13]; this maneuver helps to prevent immoderate lordosis of the lower back and forward tilt of the pelvic [14]. In AB, the lumbar spine is fixed by using isometric co-contraction of all muscles in the trunk [15]; this maneuver improves vertebral stability against sudden perturbations and reduces lumbar spine movement [16]. AH and AB reportedly suppress spinal movements during lower limb exercises [17,18,19,20,21]. Studies comparing AH and AB have also investigated the effects of lumbar spinal movement on sudden rearward or posteroanterior loads applied to the spinous processes [16,22,23].

To the best of our knowledge, no studies have investigated the effects of AH or AB on trunk stabilization during LPF. Furthermore, the effects of AH and AB on the angle of pelvic rotation during LPF have not been studied. Thus, this study was performed to explore the effects of AH and AB on electromyography (EMG) activity of the trunk muscles and angle of pelvic rotation during LPF in healthy volunteers. We hypothesized that each condition (AH, AB, and without any condition) during LPF exercise would have a different effect on muscle activation and pelvic rotation angle. 

## 2. Methods 

### 2.1. Subjects

Twenty healthy adult females (mean age = 27.53 ± 5.97 years; mean height = 163.27 ± 5.24 cm; mean weight = 48.67 ± 6.28 kg) participated in this study. Inclusion criteria were the age of 18–65 years and the absence of current low back pain. Exclusion criteria were any of the following: diagnosis of neurological or joint disease, known abdominal or respiratory pathology, history of abdominal or spinal surgery, or body mass index > 30 kg/m^2^ [24]. Before the study began, the principal investigator explained all procedures to the subjects. Ethical approval for this study was obtained from the Institutional Review Board of Inje University in the Republic of Korea (INJE 2022-07-006); written informed consent was obtained from all subjects before the beginning of data collection. The sample size was estimated using G*Power 3.1.9.7 for Windows (G*Power©, University of Dusseldorf, Dusseldorf, Germany). Power analysis indicated that at least 17 subjects were needed to achieve a power of 0.80 at a significance level of *p* < 0.05. This analysis was based on an estimated effect size derived from the previous literature [25]. 

### 2.2. Surface Electromyography and Data Processing

A Delsys Trigno Wireless electromyography (EMG) system (Delsys Inc., Boston, MA, USA) was used to assess the EMG activity of the bilateral TrA/IO, EO, and MF muscles. The EMG sensor was used for surface EMG; analog signals recorded from each muscle were converted to digital signals and processed by DELSYS EMG Works acquisition software on a personal computer. The sampling rate was 2000 Hz, with a 20~450 Hz bandpass filter. All EMG data were full-wave rectified and converted to root-mean-square (RMS) values using a 125 ms window. The area for each electrode was shaved, and then cleaned with cotton and alcohol to reduce skin impedance. For collection of bilateral TrA/IO activity data, EMG sensors were attached approximately 2 cm medial and lower to the anterior superior iliac spines (ASIS) [26]. For the bilateral obliquus externus abdominis (EO), electrodes were attached inferolateral to the 8th rib [27]. Finally, EMG sensors were attached 2 cm lateral from the L5 spinous process to record bilateral multifidus (MF) activity [28] (Figure 1).

The maximum voluntary isometric contraction (MVIC) values were measured through manual muscle testing to normalize the EMG values for TrA/IO, EO, and MF muscles [29]. The MVIC trials for each muscle were performed twice with a rest period of 1 min between trials. The mean EMG value of the middle 3 s of MVIC trials was used to normalize values for each muscle. All EMG data during LPF exercises are expressed as percentages of MVIC (%MVIC). 

### 2.3. Smart Phone-Based Measurement Tool

For measurement of transverse rotation on pelvis, a smart phone was connected to a smart phone holder designed to work with the Smartphone-Based Measurement Tool (SBMT). The SBMT frame was hand-made using a wooden bar and a smart phone holder. The SBMT contains a wooden frame that holds a smart phone in a position where it can measure the transverse pelvic rotation angle. A previous study demonstrated that measurement of pelvic rotation using the SBMT has excellent reliability [30]. Before pelvic rotation measurements, the SBMT was placed on a flat surface to calibrate the Apple inclinometer application (Clinometer level and slope finder; Plaincode Software Solutions, Stephanskirchen, Germany). Then, the SBMT was placed on both ASIS. The Apple inclinometer application was used to record the angle of pelvic rotation during the LPF. 

### 2.4. Pilates Exercises 

Each subject performed the LPF exercises (Figure 2). A Pilates instructor who had received training in the Pilates method instructed the subjects in LPF performance for approximately 5 min. To begin, subjects assumed a plank position (Figure 2); they were then asked to extend the hip of the dominant lower extremity to a predetermined target bar. The Pilates principles of body alignment, breathing control, and abdominal muscle control were emphasized throughout the exercise session. The exercise was performed three times for 5 s each with a 1 min rest period between trials.

### 2.5. Experimental Procedures

Subjects were asked to gently draw the lower abdominal wall in toward the spine for AH; they were asked to tighten their abdominal wall and increase the lateral diameter of the waist for AB. Prior to the experiment, subjects were provided with a complete explanation of AH and AB; they practiced each maneuver for 5 min each to ensure that they could achieve a satisfactory result. LPF with AH (LPF-AH), LPF with AB (LPF-AB), and LPF without any condition (LPF-WC) were performed randomly (www.randomlists.com, accessed on 19 December 2022). To prevent muscle fatigue and learning effects, subjects rested for 5 min between conditions [31].

First, subjects were asked to assume a plank position. A goniometer was used to set each subject’s hip extension angle at 10°, and a target bar was set. The dominant leg was distinguished by asking the subject to kick a soccer ball; the kicking leg was regarded as the dominant leg [32]. All subjects were right-leg dominant. To measure the angle of pelvic rotation, LPF-AH, LPF-AB, and LPF-WC were conducted with the hip extended to the point where the target bar was reached (Figure 2a). In total, the 3 times exercise was performed; the mean value of the result was used for subsequent data analysis. For measurement of EMG amplitude, subjects conducted with three conditions and then performed hip extension for 5 s, such that the ankle hit the target bar (Figure 2b). The measurements were repeated 3 times. EMG signals were collected only 3 s in the middle 5 s, and the first and last second were excluded; the mean value for each EMG signal was used for subsequent data analysis. All measurements were conducted with the trunk, hips, and legs maintained in a straight orientation.

### 2.6. Statistical Analysis

ANOVA was used to compare differences in the pelvic rotation angle and the EMG amplitudes of the TrA/IO, EO, and MF among the three conditions (LPF-AH, LPF-AB, and LPF-WC). The ANOVA were followed by post hoc comparisons with Bonferroni correction. All data were analyzed using PASW software (ver. 18.0; SPSS Inc., Chicago, IL, USA). Statistical significance was set at *p* < 0.05.

## 3. Results

### 3.1. Electromyographic Activity of the Trunk Muscles

The results of EMG activities are summarized in Table 1. Of the three conditions, significant differences in the EMG activities of each muscle were observed. The muscle activities of the Rt. TrA/IO were significantly greater for LPF-AH (58.46 ± 26.7) compared to those of LPF-AB (42.8 ± 20.39) and LPF-WC (27.23 ± 14.05). The muscle activities of the Rt. TrA/IO for LPF-AB were significantly greater than those of LPF-WC (*p* < 0.001). The muscle activities of the Lt. TrA/IO were significantly greater for LPF-AH (52.9 ± 24.47) compared to those of LPF-AB (45.88 ± 19.15) and LPF-WC (34.56 ± 18.86). The muscle activities of the Lt. TrA/IO for LPF-AB were significantly greater than those of LPF-WC (*p* = 0.002).

The muscle activities of the Rt. MF were significantly greater for LPF-AB (42.47 ± 17.78) and LPF-AH (40.48 ± 21.03) compared to those of LPF-WC (31.29 ± 13.06). The muscle activities of the Rt. MF were not statistically significant in LPF-AB and LPF-AH (*p* = 1.000). The muscle activities of the Lt. MF were significantly greater for LPF-AB (28.55 ± 12.2) and LPF-AH (23 ± 7.51) compared to those of LPF-WC (15.93 ± 4.45). The muscle activities of the Lt. MF were not statistically significant in LPF-AB and LPF-AH (*p* = 0.098).

The muscle activities of the Rt. EO was significantly greater for LPF-AH (42.57 ± 15.88) compared to those of LPF-WC (37.29 ± 14.27). The muscle activities of the Rt. EO were not statistically significant in LPF-AH and LPF-AB (*p* = 0.310). The muscle activities of the Lt. EO were not statistically significant under three conditions (LPF-AH, 40.76 ± 19.42; LPF-AB, 41 ± 17.52; LPF-WC, 37.32 ± 16.73).

### 3.2. Angle of Pelvic Rotation

The pelvic rotation angle was observed under three conditions (LPF-AH, 3.78 ± 2.45°; LPF-AB, 2.15 ± 2.03°; LPF-WC, 6.49 ± 3.66°) (Figure 3). These angles significantly differed among the three conditions (*p* < 0.001). Post hoc analysis showed that the pelvic rotation angle was significantly smaller in LPF-AB than in LPF-AH (*p* < 0.001) or LPF-WC (*p* < 0.001). Furthermore, the angle of pelvic rotation was significantly smaller in LPF-AH than in LPF-WC (*p* = 0.004).

## 4. Discussion

This study was performed to investigate the angle of pelvic rotation and the muscle activities of the TrA/IO, EO, and MF in three LPF conditions. We found significant increases in the muscle activities of the TrA/IO and right EO during LPF with AH compared to those of LPF with AB. In addition, we observed a decrease in pelvic rotation during LPF with AB compared to those of LPF with AH.

The bilateral TrA/IO activity was significantly greatest in AH among all conditions, probably because contractile strategies differ between AH and AB. The AH maneuver is presumed to be a preferential contraction exercise for activating deep muscles such as the TrA, IO, lumbar multifidus, and diaphragm [1,33,34]. In contrast, the AB maneuver does not focus on specific muscle recruitment; it involves contractions of all abdominal and low back muscles, including deep muscles such as the TrA and lumbar multifidus [16,34,35,36]. In AB, co-contraction of all abdominal muscles does not ensure equal contraction among muscles. Therefore, muscle activity is expected to be greater in the AH maneuver, which selectively contracts the TrA, than in the AB maneuver. AH is generally considered more effective for the enhancement of TrA activity during the LPF.

In the present study, there is no significant difference between LPF-AH and LPF-AB. The reason may be that when TrA contracts, MF contracts simultaneously [37]. AB and AH are recommended to facilitate TrA activation [1,38]. In a previous study, AH increased IO muscle activity [35,38], leading to co-contraction of the MF [34,37] and enhancement of trunk stability. Previous studies have shown no significant differences between AH and AB with respect to MF activation during prone hip extension [27]. Because these muscles attach to the lumbar spine directly or indirectly by means of the thoracolumbar fascia, their activation increases spinal stability and reduces unwanted lumbar movement [1,7]. Therefore, MF activity during LPF was increased with both AB and AH; the difference between the two conditions was not significant.

The bilateral EO activities did not significantly differ between LPF-AH and LPF-AB. The muscle activity of the left EO did not significantly differ among any of the three conditions. Previous studies have reported that AH increases EO activity compared to the control condition [39]; our findings are consistent with the previous results. In oblique musculature, the EO would be less affected by AH than the IO; the EO is a more global torque-producing muscle that exhibits less involved in segmental spinal stability [40]. However, the medial fibers of the EO are anatomically associated with the thoracolumbar fascia; they are presumed to provide some biomechanical stability [41,42]. Thus, the IO and TrA are expected to function in a synergistic manner to flatten the abdomen during AH [43]. Additionally, the prone position has been shown to affect abdominal muscle recruitment during AH, particularly by increasing EO activity. This is possibly because of the greater gravitational demands or reflex-mediated activities of these superficial muscles in responding to stretching during AH [44,45]. In addition, moderate activation of the EO has been observed during performance of the standard prone plank [46,47,48,49]. The activity of the left EO muscle controls the postural muscles [6], while the right EO may prevent unwanted pelvic rotation during the LPF. We suspect that AH can help to reduce pelvic rotation by activating the abdominal oblique muscles to stabilize the trunk during LPF; this hypothesis is supported by the increased EO activation that we observed during LPF-AH.

In this study, the angle of pelvic rotation was significantly smaller in LPF-AB than in the other conditions. In addition, LPF-AH caused significantly less movement than LPF-WC and significantly more movement than LPF-AB. These findings suggest that, compared with AH, AB can cause greater reduction in pelvic rotation during LPF. These findings are consistent with previous studies results to show that AB is more effective than AH in maintaining lumbopelvic stability [16,35]. However, contrary to our findings, Suehiro et al. [27] compared AH and AB during prone hip extension; they reported no significant differences in thoracic, lumbar, or pelvic kinematics. Those results suggest that AH can improve the anteroposterior lumbopelvic region stabilization by the pelvic posterior tilting and reducing the lordosis of the lumbar spine. Thus, our findings imply that AB can help to decrease pelvic rotation during LPF.

Many Pilates instructors use the LPF to increase trunk muscle activity and many studies have examined the effects of the various types of plank exercises. To our knowledge, there has been a lack of research concerning the effects of the AH and AB maneuvers on specific muscle activity and pelvic rotation during LPF. Our current findings indicate that AB may reduce pelvic rotation during the LPF; moreover, they suggest that AH may cause greater enhancement of TrA/IO activity compared with AB.

This study had several limitations. First, because we focused on healthy adult women, the study findings may not be generalizable to other populations. Second, age specificity was not considered because we have a wide age range. Third, we could not confirm whether muscle activity increased bilaterally during hip extension with AH and AB because we only measured the trunk muscles. Future studies should focus on measurements of bilateral hip extensor activation. Finally, this study could not confirm the long-term effects of AH and AB during LPF because we used a cross-sectional design. 

## 5. Conclusions

This study examined the effects of AH and AB on the EMG signal amplitude of the trunk muscles and on the angle of pelvic rotation during LPF. The results suggest that AH is more effective than AB for increasing TrA/IO activity during LPF. Furthermore, AB may reduce pelvic rotation during LPF. Therefore, LPF-AH is suitable for selective activation of the TrA/IO, while LPF-AB is recommended for the prevention of unwanted pelvic rotation.

## Figures and Tables

**Figure 1 healthcare-11-00060-f001:**
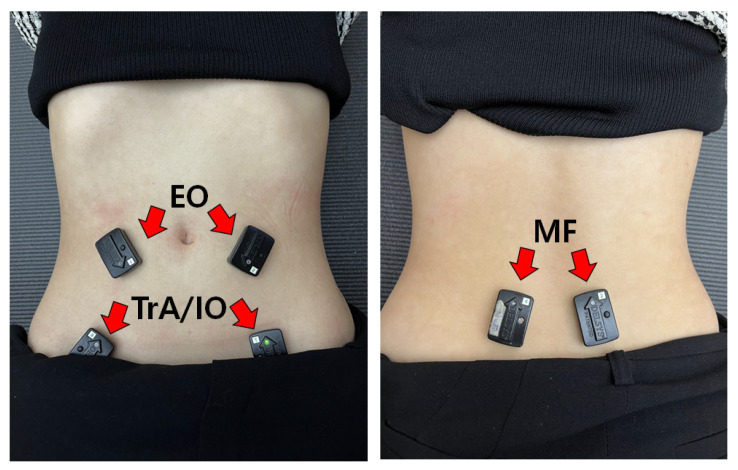
Surface Electromyography sensor attachment.

**Figure 2 healthcare-11-00060-f002:**
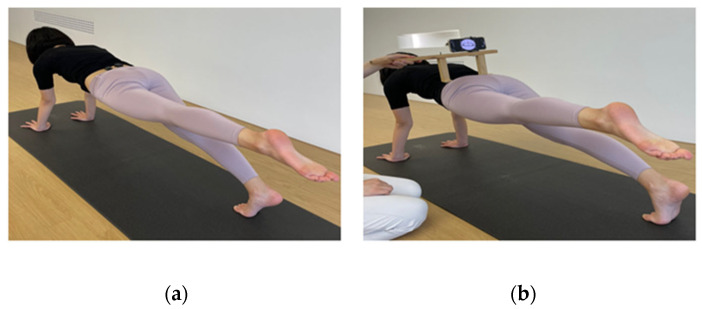
Measurement of muscle activity and pelvic rotation during LPF. (**a**) Measurement of muscle activity in the TrA/IO, EO, and MF; (**b**) measurement of pelvic rotation angle.

**Figure 3 healthcare-11-00060-f003:**
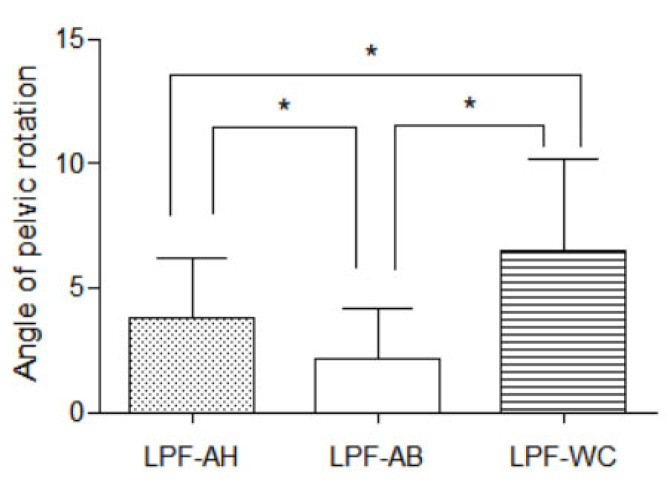
Comparison of the angle of pelvic rotation between the conditions; * *p* < 0.05; means and standard deviations are denoted by bars, respectively.

**Table 1 healthcare-11-00060-t001:** Electromyography activities of each muscle among three conditions (*n* = 20).

Muscle Activity (%MVIC)	LPF-AH	LPF-AB	LPF-WC	*f*	*p*
**Rt. TrA/IO**	58.46 ± 26.70 ^a,c^	42.80 ± 20.39 ^b^	27.23 ± 14.05	20.787	<0.001 **
**Lt. TrA/IO**	52.90 ± 24.47 ^a,c^	45.88 ± 19.15 ^b^	34.56 ± 18.86	11.949	<0.001 **
**Rt. EO**	42.57 ± 15.88 ^c^	39.89 ± 14.66	37.29 ± 14.27	5.068	0.018 *
**Lt. EO**	41.00 ± 17.52	40.76 ± 19.42	37.32 ± 16.73	1.615	0.226
**Rt. MF**	40.48 ± 21.03 ^c^	42.47 ± 17.78 ^b^	31.29 ± 13.06	18.302	<0.001 **
**Lt. MF**	23.00 ± 7.51 ^c^	28.55 ± 12.20 ^b^	15.93 ± 4.45	23.181	<0.001 **

Values are presented as means ± standard deviations; * *p* < 0.05, ** *p* < 0.001; MVIC = maximal voluntary isometric contraction; LPF-AH = leg pull front with abdominal hollowing; LPF-AB = leg pull front with abdominal bracing; LPF-WC = leg pull front without any condition; (a) significant differences between LPF-AH and LPF-AB conditions; (b) significant differences between LPF-AB and LPF-WC conditions; (c) significant differences between LPF-AH and LPF-WC conditions.

## Data Availability

Not applicable.

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
