# Peer review of "The Effects of Abdominal Hollowing and Bracing Maneuvers on Trunk Muscle Activity and Pelvic Rotation Angle during Leg Pull Front Pilates Exercise"

_healthcare, 2022, doi:10.3390/healthcare11010060_

Round 1

Reviewer 1 Report

Overall: the study could be considered a pilot study, on the effect of two exercise on muscle activity and pelvic rotation. Is not clear enough what is the research hypothesis and what are the novelty of the finding.
Title: quite clear and concise.
Abstract: English and typo edit are needed; moreover, abstract was not so clear and did not report clearly the main results of the study.
Introduction: lack the study hypothesis
Methods: age range is really wide (this is not mentioned in the limitation).
Sample size calculation need more details.
Why EMG data were root converted, probably a clearer motivation must be given.
Pay attention to typos cot-ton (lines 87) con-ducted (lines 134)
To understand position, movement, and electrodes position probably Authors need to show some more photos or figures.
Lines 124: the exercise sequence was administered randomly, is it correct? If so, explain it better.
The smart phone holder was handmade, stated please.
Results
Lines 147 and 174 “it shows the result” try a different incipit for the sentence.
I risultati sono oscuri per me. Non capisco cosa vogliono dire, forse tu riesci a capire.
Lines 158; differences were not statistically significant, is more appropriate.
In the model the sequence of exercise administration must be considered in the model (to verify a carryover effect)
Post hoc analysis by means of what test and referred to what?
Discussion
Discussion needs to be ameliorated. Authors substantially repeat the results presentation. This is the consequence of the lack of a clear hypothesis.

Reviewer 2 Report

In general the study is well written. I have few comments.

Abstract. 

Line 22-23. Therefore, LPF-AH is suitable for selective activation of the TrA/IO, while LPF-AB is recommended for the prevention of unwanted pelvic rotation. Put in the context of trunk muscle stabilization exercises.

Introduction

Line 28-33. First paragraph is lack of direction Provide an overview of the whole study.

Methods

Line 88-92. Provide a figure showing the electrode placements.

Line 93. Provide information on familiarization. How many contractions (MVIC) were needed for the EMG to stabilize during familiarization? 
